# *Thunbergia laurifolia* Leaf Extract Increased Levels of Antioxidant Enzymes and Protected Human Cell-Lines In Vitro Against Cadmium

**DOI:** 10.3390/antiox9010047

**Published:** 2020-01-06

**Authors:** Marasri Junsi, Chutha Takahashi Yupanqui, Worapong Usawakesmanee, Alan Slusarenko, Sunisa Siripongvutikorn

**Affiliations:** 1Department of Food Technology, Faculty of Agro-Industry, Prince of Songkla University, Hat-Yai, Songkhla 90112, Thailand; marasri_jun@hotmail.com (M.J.); worapong.u@psu.ac.th (W.U.); 2Interdisciplinary Graduate School of Nutraceutical and Functional Food, Prince of Songkla University, Hat-Yai, Songkhla 90112, Thailand; chutha.s@psu.ac.th; 3Department of Plant Physiology, RWTH Aachen University, 52056 Aachen, Germany; alan.slusarenko@bio3.rwth-aachen.de

**Keywords:** *Thunbergia laurifolia*, anti-cadmium, antioxidant enzymes, HEK293 cell, HepG2 cell

## Abstract

*Thunbergia laurifolia* or Rang Jued has been used as an herbal tea and in folk medicine as a detoxifying agent. Cd contamination is globally widespread and a serious public health problem. The aim of this study was to determine the endogenous antioxidant enzyme activities and malondialdehyde (MDA) production of the crude dried extract (CDE) of *T. laurifolia* leaves, using human embryonic kidney (HEK293) and human liver (HepG2) cells as in vitro models. Moreover, the cytotoxicity including anti-cadmium (Cd) toxicity in both cells were measured. The experimental design had 3 treatment groups with combined, pre-, and post-treatments for investigating the anti-Cd toxicity, and cell viability was determined with MTT test (3-(4,5-dimethylthiazol-2-yl)-2,5-diphenyltetrazolium bromide). The CDE showed low cytotoxicity and increased catalase (CAT) and glutathione peroxidase (GPx) activities with decreased malondialdehyde (MDA) levels in both cell types. It was found that the CDE protected against Cd-induced toxicity in both cell types, and a synergistic combination therapy effect was seen when CaNa_2_EDTA, a chelating agent, was applied. Therefore, CDE can protect against Cd-induced oxidative stress in cells, possibly due to its antioxidant properties. Moreover, using the extract or drinking the herbal tea together with chelating agent should have an efficacy advantage over using the CDE or the chelating agent singly.

## 1. Introduction

Cadmium (Cd) is a hazardous chemical element to most living cells, particularly those of the higher animals and humans. Its half-life in the human body is about 30 years, which is obviously quite long, and it is difficult to remove [1]. Cd residues cause serious international concerns due to bioaccumulation effects on human health [2]. Accumulation of Cd can be found in almost all types of animal tissues, especially kidney as it is a major target organ [3,4]. Numerous scientific studies have shown that Cd is harmful through various mechanisms, involving disruption of cell adhesion, disruption in cellular signal transduction and apoptosis, inhibition of DNA repair including activity of cellular antioxidant enzymes leading to the loss of balancing between antioxidant enzyme and reactive oxygen species (ROS) generation and result in oxidative stress in cell [5,6,7]. Oxidative stress is a condition associated with an increased rate of cellular damage induced by the ROS and oxidized active molecules (O_2_, O_2_•^−^, OH•, H_2_O_2_). Lipid peroxidation products, especially malondialdehyde (MDA) [1] lead to loss of balance between ROS generation and scavenging activities. Actually, in a normal and healthy body there is a balance between reactive oxygen species formation or free radical and endogenous antioxidant defense mechanisms [8]. In addition, the ROS are mostly removed by endogenous antioxidants enzymes, including catalase (CAT) and glutathione peroxidase (GPx) [9,10]. 

*Thunbergia laurifolia* or Rang Jued, a local Thai plant belonging to the family Acanthaceae, is commonly consumed as an herbal tea or even as a fresh vegetable side dish. Fresh and dried plant leaves, bark and roots are used traditionally as antidotes for insecticides, drugs, heavy metals, and toxic chemical exposure [11,12]. Rang Jued leaves have also been reported as sources of bioactive compounds and have been utilized as antioxidants [13,14]. In addition, our previous studies showed that the CDE had a high total extractable phenolic content (TPC) and a high antioxidant activity based on radical scavenging, 2,2-diphenyl-1-picrylhydrazyl (DPPH), 2,2’-azino-bis(3-ethylbenzothiazoline-6-sulphonic acid) (ABTS) and ferric ion reducing antioxidant power assay (FRAP) [15]. It is known that in the human body liver and kidney organs deal with elimination of toxic metabolites and are major target sites of Cd-induced acute and chronic toxicity [16,17]. Therefore, the objective of this work was to investigate the biological activities of CDE from *T. laurifolia* leaves determined by cytotoxicity, endogenous antioxidant enzyme activities (CAT and GPx) and MDA production, including the influence of the extract on anti-Cd toxicity in HEK293 kidney and HepG2 liver cells.

## 2. Materials and Methods 

### 2.1. Chemicals and Reagents

Chemicals used for cells cultures were purchased from Gibco (Carlsbad, California). Most of the chemicals used for antioxidant activity determinations were purchased from Sigma-Aldrich, Seelze, Germany; otherwise from Merck, Darmstadt, Germany; Ajax Finechem, Auckland, New Zealand; QRAC, Selangor, Malaysia; Fisher Scientific, Leicestershire, England; and LAB-SCAN, Dublin, Ireland. 

### 2.2. Raw Material

*T. laurifolia* leaves collected at a developing or an intermediate stage (between dark green and bright green color, and without variegation) which can be folded without brittle breaking were directly purchased from a farmer (Bangkhen, BKK, Thailand), and transported to the laboratory within 24 h.

### 2.3. Plant Preparation and Extraction

*T. laurifolia* leaves were removed from climber, washed with tap water, drained and air dried, (following a folk medicine method) for 5–8 d to a moisture content of 8–10% (*w/w*), ground to a fine powder passing through 20–40 mesh, and stored in dark bottles at room temperature up to use within 6 mo. A powder sample (2 g) was soaked in 20 mL of hot water, at 95 ± 2 °C (1:10 *w/v*) for 1 h to obtain the crude extract (300 mg), the original concentration of a beverage tea, and then filtered through three layers of gauze followed by Whatman No. 4 filter paper. The filtrate was freeze-dried and stored at 4 °C for further study as a crude dried extract (CDE) [18].

### 2.4. Proximate Compositions and Mineral Contents

The powder of *T. laurifolia* was analyzed for proximate composition in terms of protein, ash, fat, fiber, carbohydrate and mineral contents (magnesium, copper, zinc, and iron) according to the method in [19].

### 2.5. Cell Culture 

Human embryonic kidney cells (HEK293) were purchased from American Type Culture Collection (Manassas, VA, USA) and grown in minimum essential medium (MEM), while human liver cells (HepG2) were kindly provided by Teerapol Srichana (Faculty of Pharmaceutical Sciences, Prince of Songkla University, Thailand) and were cultured in Dulbecco’s Minimal Essential Media (DMEM). Both media were supplemented with 10% fetal bovine serum (FBS) and 1% penicillin-streptomycin. The cells were culture at 37 °C in a humidified atmosphere of 5% CO_2_ and 95% air in a fully humidified incubator. The cells were harvested with 0.25% trypsin-EDTA and suspended in a fresh medium. Cells were counted by a standard trypan blue cell counting technique [20].

### 2.6. Determination of the CDE Effects on Antioxidant Enzyme Activity and Malondialdehyde Value

#### 2.6.1. Preparation of Endogenous Cellular Extract

All cell lines were seeded at 1 × 10^6^ cells/mL in 60 mm tissue culture dishes and incubated for 24 h to allow the cells to grow and adhere to the dish. In each culture dish, the cells were washed twice with 2 mL of PBS (pH 7.2) before treating with the CDE solutions (0.01–1.00 mg/mL). The CDE solutions were added to the dishes and they were incubated for 24 h at 37 °C before removal from cell culturing. Cells in each culture dish were harvested by incubation (1 min for HEK293 and 5 min for HepG2) with 0.5 mL of 0.25% trypsin-EDTA. Then, 1 mL of culture media was added and centrifuged at 1000× *g* for 10 min. Cell pellets were washed with ice chilled PBS (500 mL, 2 times). Cell pellets were then lysed by sonication on ice for 1 min using a probe-type sonicator (Vibra-Cell, Sonics and Materials Inc., Newtown, CT, USA) pulsing at 15 s on and 10 s off cycle [21]. The mixture was then centrifuged at 10,000× *g* for 10 min at 4°C and the supernatant (endogenous cellular extract, ECE) was assayed for enzyme activity. Protein contents were determined with a modified Bradford‘s assay [22] using bovine serum albumin as a standard for protein concentration.

#### 2.6.2. Catalase (CAT) Activity

The activity of catalase (CAT) was determined by monitoring the decrease in absorbance at 240 nm due to hydrogen peroxide (H_2_O_2_) consumption, following the method of [23] with minor modifications. The 1.9 mL reaction mixture with 50 mM phosphate buffer (pH 7.0) and 0.25 mM H_2_O_2_ was mixed with 100 µL ECE in a UV cuvette and an ultraviolet spectrophotometer measured the absorbance.

#### 2.6.3. Glutathione Peroxidase (GPx) Activity

The activity of glutathione peroxidase (GPx) was determined based on glutathione oxidation by GPx in the presence of 5,5′-dithiobis(2-nitrobenzoic acid) (DTNB) following the method of [24] with minor modifications. Briefly, 200 μL of ECE was added to 400 μL of 0.1 mM glutathione (GSH), 200 μL of 0.067 M sodium hydrogen phosphate (Na_2_HPO_4_), then incubated at 28 °C for 5 min before adding 200 μL of 1.3 mM H_2_O_2_, and stored at room temperature for 10 min. This was followed by adding 1 mL of 1% trichloroacetic acid (TCA) and storing in ice bath for 30 min. Then, 480 μL of the mentioned reaction mixture was added to 2.2 mL of 0.32 M Na_2_HPO_4_, 320 μL of 1 mM DTNB and stored at room temperature before measuring the absorbance at 412 nm. Both enzyme activities were calculated as enzyme units per mg protein, according to the calibration curve of standard bovine serum albumin (BSA) by Bradford‘s assay [22], and are expressed as units per mg protein and percentage of the control.

#### 2.6.4. Malondialdehyde (MDA) Value

To measure the malondialdehyde (MDA) value, the modified method in [25] was used. First 600 μL of ECE were mixed with 3 mL of 20% TCA containing 0.8% of thiobarbituric acid (TBA) (*w/v*) and heated at 95 °C for 60 min. The reaction mix was cooled down under running tap water and centrifuged at 3000× *g* for 15 min at 4 °C before measuring the absorbance at 532 nm. The content of MDA was analyzed following the calibration curve for standard MDA, and is expressed as nmol per mg protein and percentage of the control.

### 2.7. Determination of CDE Cytotoxicity 

The cells were seeded in a 96-well plate at 1 × 10^6^ cells/mL by adjusting the cell density with culture medium and counting with standard trypan blue cell counting technique, before incubation for 24 h to allow cells to adhere to the flask. The CDE at concentrations of 0.01–2.00 mg/mL were added to the wells and incubated for 24 h. The solution was removed from the cell cultures and then the cells were treated with 3-(4,5-dimethylthiazolyl-2)-2,5-diphenyltetrazolium bromide (MTT) [26]. Briefly, 100 µL of MTT (0.5 mg/mL) was added to each well and incubated for 3 h at 37 °C. After incubation, the media was aspirated and 100 μL of DMSO was added to each well to dissolve the formazan. Cells were incubated for 10 min at 37 °C before reading the absorbance at 570 nm with a microplate spectrophotometer. To determine the effects of CDE on MTT assays, the experiment was set up as follows.
Normal control:   Media + Media
Tested sample:   Media + CDE

The cell viability was calculated to percentage of cell viability as follows.
% Cell viability = [Absorbance of sample/Absorbance of control] × 100(1)

### 2.8. Determination of CDE’s Anti-Cd Property

Cytotoxicity of Cd was tested on HEK293 and HepG2 cells by the MTT assay [26]. The concentration at 50% of cytotoxicity (CC_50_) of Cd agent was found from a plot of the percentage cells survival versus the concentration of Cd. Briefly, cells were plated in a 96-well plate at a density of 1 × 10^6^ cells/mL (counted by a standard trypan blue cell counting technique) and incubated for 3 h at 37 °C, to allow cells to attach before treatment with the Cd solution at various concentrations (20–120 µmol/L) with incubation for 24 h at 37 °C. Then, 100 µl of MTT (0.5 mg/mL) was added to each well with incubation for 3 h at 37°C. After incubation, the media were aspirated and 100 μL of DMSO was added to each well to dissolve the formazan for 10 min at 37 °C before reading the absorbance at 570 nm with a microplate spectrophotometer. The effects of CDE on each cell type when induced by Cd at CC_50_ was investigated. Briefly, cells were plated in a 96-well plate at a density of 1 × 10^6^ cells/mL and allowed to attach to the culture plate (24 h) before treating with CDE or the chelating agent. There were three treatment groups: together (combined-treatment), before (pre-treatment), and after treatment with Cd at CC_50_ (post-treatment). In this experiment, calcium disodium ethylenediamine tetraacetic acid (CaNa_2_EDTA) was used as the chelating agent and a positive control. The treatments were as follows. 

Group 1 (combined-treatment by adding the extract/CaNa_2_EDTA and Cd (CC_50_) together)
Normal control: Media (24 h) CC_50_ of Cd (negative control): Media and Cd at CC_50_ (24 h)Tested sample: CDE/CaNa_2_EDTA and Cd at CC_50_ (24 h)

Group 2 (Pre-treatment by adding the extract/CaNa_2_EDTA before Cd (CC_50_))
Normal control: Media (24 h) + Media (24 h)CC_50_ of Cd (negative control): Media (24 h) + Cd at CC_50_ (24 h)Tested sample: CDE (24 h)/CaNa_2_EDTA + Cd at CC_50_ (24 h)

Group 3 (Post-treatment by adding the extract/CaNa_2_EDTA after Cd (CC_50_))
Normal control: Media (24 h) + Media (24 h)CC_50_ of Cd (negative control): Cd at CC_50_ (24 h) + Media (24 h)Tested sample: Cd at CC_50_ (24 h) + CDE/CaNa_2_EDTA (24 h)

The cell viability was again detected by MTT cytotoxicity assay and percentage cell viability was calculated from Equation (1). 

### 2.9. CDE Coupled with a Chelating Agent (CaNa_2_EDTA) as Anti-Cd Treatment

The combination of CDE and CaNa_2_EDTA was tested against Cd toxicity on HEK293 and HepG2 cells and the MTT assay. The groups and treatments were as follows. 

Group 1 (Pre-treatment by adding the extract before CaNa_2_EDTA mixed with Cd (CC_50_))
Normal control: Media (24 h) + Media (24 h)CC_50_ of Cd (negative control): Media (24 h) + Cd at CC_50_ (24 h)CC_50_ of Cd + EDTA (positive control): Media (24 h) + CaNa_2_EDTA mixed with Cd at CC_50_ (24 h)Tested sample: CDE (24 h) + CaNa_2_EDTA mixed with Cd at CC_50_ (24 h)

Group 2 (Post-treatment by adding the extract after CaNa_2_EDTA mixed with Cd (CC_50_))
Normal control: Media (24 h) + Media (24 h)CC_50_ of Cd (negative control): Cd at CC_50_ (24 h) + Media (24 h)CC_50_ of Cd + EDTA (positive control): CaNa_2_EDTA mixed with Cd at CC_50_ (24 h) + Media (24 h)Tested sample: CaNa_2_EDTA mixed with Cd at CC_50_ (24 h) + CDE (24 h)

The equation for a calculated percentage of cell viability as followed Equation (1). Morphology of HEK293 and HepG2 cells treated with Cd-induced toxicity and crude extract coupled with CaNa_2_EDTA were then measured by microscope at 40× (Nikon eclipse TS100, Nikon Corporation, Minato-ku, TK, Japan).

### 2.10. Statistical Analysis

The experimental design was completely randomized (CRD). Data were subjected to analysis of variance (ANOVA). Post hoc comparisons of means were carried out by Duncan’s multiple range tests. Significance was declared for *p* < 0.05 using SPSS (SPSS Inc., Chicago, IL, USA).

## 3. Results and Discussion

### 3.1. Proximate Composition and Mineral Contents in T. laurifolia Leaves

The proximate compositions and contents of several minerals in dried leaves of *T. laurifolia* are shown in Table 1. The results show that crude protein, crude fat, crude fiber, ash and carbohydrate contents in the dried leaves are 13.98 ± 1.90%, 1.83 ± 0.12%, 11.16 ± 0.44%, 19.93 ± 1.90 and 53.10 ± 2.23% (dry weight basis), respectively. The prior study [27] reported that the *T. laurifolia* leaves contained crude protein, crude fat, crude fiber, ash, and carbohydrate as 16.70%, 1.68%, 16.82%, 18.79%, and 46.01% based on dry basis weight, respectively. Regarding the proximate composition, it was found that the dried leaves of this study have very high ash content compared with commercial black tea (4.94 ± 0.70%) and green tea (4.57 ± 0.82%) [28]. Besides the type and variety of plant, the chemical composition and mineral contents in leaves may differ as a result of several factors, such as the maturity stages of leaves and plant, growth environment including local soil composition [29,30]. 

The contents of mineral in *T. laurifolia* leaves indicate them as a good source of various micronutrients, such as potassium (K), phosphorus (P), and magnesium (Mg) (Table 1), suited for some purposes such as supplements, animal feed, fertilizer, and so on. Mg is one of the most important elements involved in many enzyme activities and the structural stabilization of tissues [31]. This element is useful to carbohydrate metabolism and to cell membrane stabilization in plants [32]. P is required for ATP and nucleic acid synthesis (RNA and DNA), and in protein production, while K has a profound effect on the profile and distribution of the primary metabolites in plant tissues such as organic acids, amino acids and soluble sugars, particularly reducing sugars. Moreover, K is reported to be important cofactor in several enzyme systems, such as Na^+^/K^+^-ATPase in Na^+^- K^+^ pump system to keep homeostasis of the animal cells, and pyruvate kinase, which is an important enzyme in carbohydrate metabolism in plant cells [33,34]. 

The microelement contents such as zinc (Zn), copper (Cu), iron (Fe) and selenium (Se) of *T. laurifolia* leaves were ranked as follows: Fe > Zn > Cu > Se. Microelements are required in small concentrations and are important for growth and other plant functions. Fe is an essential microelement for almost all living organisms because it plays a critical role in numerous metabolic processes, such as DNA synthesis, respiration, and photosynthesis [35]. Furthermore, Zn is an essential element to all organisms, particularly in the form of Zn^2+^. It acts as a catalytic or co-factor for numerous enzymes including the chelator nicotianamine, a non-proteinogenic amino acid acting as a high-affinity metal chelator in the plant for homeostasis balancing [36]. Moreover, Zn is a most relevant mineral to human health, because of its antioxidant and anti-inflammatory properties. [37] suggested that Zn acts as a cofactor for important enzymes involved in the proper functioning of the antioxidant defense system, protects cells against oxidative damage and the stabilization of membranes by inhibiting the pro-oxidant enzyme, nicotinamide adenine dinucleotide phosphate oxidase (NADPH-Oxidase). Cu and Se are also required for many enzyme activities in plants and has influenced the vigor of plants and leaves including antioxidant balancing to offer the possibility of disease management [38,39,40]. These trace elements are essential components in antioxidant enzymes for many organisms, including plants, animals, and humans. For example, glutathione peroxidase (GPx), cytoplasmic superoxide dismutase (SOD) and catalase (CAT) enzymes, require Se, Cu-Zn, and Fe metals as cofactors, respectively [41]. Therefore, it was hypothesized that these minerals would interact with biological activities, including antioxidant enzyme activity.

### 3.2. Antioxidant Enzyme Activities and Malondialdehyde Values in the Test Cells

The effects of CDE on antioxidant enzyme activities (CAT and GPx) and lipid oxidation products including MDA level in HEK293 and HepG2 cells are shown in Figure 1. In fact, CAT has its highest activity levels in liver, kidney, and red blood cells [42]. This enzyme is the first antioxidant enzyme to convert hydrogen peroxide into water and oxygen, and its catalytic reaction is a one-step process, while GPx is the main scavenger of hydrogen peroxide by catalyzing oxidation of glutathione with hydroperoxides as substrates [9]. The results show that the activities of CAT and GPx in both of cell types (Figure 1A,B) were higher than in control. CAT and GPx are intracellular enzymes, which form the primary defense system against oxidative stress. In fact, both these antioxidant enzymes are believed to play important roles in the prevention of oxidative stress-related diseases, such as cancer and cardiovascular disease, including protective effect against heavy metal-induced oxidative stress [43,44,45] reported that Cd-induced oxidative stress in rats affected the antioxidant enzymes, which needed to work to balance the oxidative stress system in cells, like CuZnSOD, CAT, GPx, glutathione reductase (GR) and glutathione-*S*-transferase (GSTs). Prior studies [15,46] have stated that the CDE of *T. laurifolia* leaves showed an array of phenolic and flavonoid compounds, including caffeic acid, rutin, isoquercetin, rosmarinic acid, catechin, quercetin, and apigenin. Generally, phenolic compounds have been claimed to be natural antioxidants scavenging free radicals and reducing lipid peroxidation. Caffeic acid was reported to improve the redox balance in the liver of rat by raising the activity of antioxidant enzymes, including GPx and CAT facilitating the removal of peroxide [47]. [48] found that caffeic acid can increase the activities of antioxidant enzymes against oxytetracycline induced hepatotoxicity in rats via the antioxidant defense system, through enhanced scavenging of free radicals. Moreover, quercetin was reported to improve antioxidant enzyme functions, including CAT and GPx in the renal tissue [49] and apigenin also up-regulated the gene expression of antioxidant enzymes [50]. In addition, our current results demonstrate that the dried leaves contain many essential elements (Fe, Zn, Cu, and Se) which serve as co-factors of the antioxidant enzymes, including CAT and GPx. Then, using fine powder of the dried leaves soaked in elevated temperature water (95 ± 2 °C for 1 h) may help extract and release more essential minerals and active compounds, which enhances cell proliferation and antioxidant effects. 

The level of MDA in this experiment is shown in Figure 1C. Generally, MDA is an end product of lipid peroxidation and is widely accepted as indicator of cell damage leading to disease onset, including Alzheimer’s disease, rheumatic arthritis, and cancer [9]. The results show that MDA level significantly decreased in HEK293 and HepG2 cell lines when the concentration of CDE increased, which is related to increased CAT and GPx activities. Earlier studies [51,52] have stated that both CAT and GPx enzymes are involved in hydrogen peroxide elimination, which is directly related to MDA reduction. The reduction of MDA level when treated with CDE may indicate oxidative stress protection. This is in agreement with Abdallah et al. [53] reporting that the combination of caffeic acid and quercetin pretreatments significantly reduced the levels of MDA from Lambda-cyhalothrin, an insecticide toxin inducing oxidative stress in rat erythrocytes. However, GSH and ROS contents should be further determined to get a solid explanation.

### 3.3. Cytotoxicity of CDE against the Test Cells

To assess the potential toxicity of CDE, a cytotoxicity assay needed to be performed, and the results are shown in Figure 2. All concentrations of CDE gave better than 80% cell viability, which is labeled as “low cytotoxicity” [13,54]. This is in agreement with the findings in [55] namely that the aqueous extract of *T. laurifolia* had only low cytotoxicity with CC_50_ > 100 µg/mL in mouse connective tissue (L929), baby hamster Syrian kidney (BHK(21)C13), human liver (HepG2) and human colon (Caco-2) cell lines. Moreover, [56] suggested that using aqueous extracts at doses ranging from 20 to 2000 mg/kg/day did not influence body weight, food consumption, behavior or general health of Wistar rats. 

However, close focus on cell viability or cytotoxicity against HEK293 and HepG2 cells showed some changes in a concentration dependent manner. All living cells tend to adapt to the environment by homeostasis, reacting to perturbations in a small range, to keep their balance [57]. Moreover, the dried leaves contained some minerals, such as K, P, and Mg, which may play important roles in a large number of cellular processes, including as cofactors in enzymatic reactions and transmembrane ion movements, especially in kidney and liver cells [58,59,60]. Therefore, these minerals in CDE may help cell viability of HEK293 kidney and HepG2 liver cells, in CDE concentration dependent manner. 

### 3.4. Cytotoxicity of Cd against the Test Cells

Cytotoxicity of the CdCl_2_ against HEK293 and HepG2 cell lines is presented in Table 2. It was found the CC_50_ of CdCl_2_ was 64.09 and 75.37 µmol/L for HEK293 and HepG2 cells, respectively. That the HepG2 cells had better resistance to CdCl_2_ may due to the liver cells having high contents of antioxidant enzymes, such as glutathione (GHS), glutathione peroxidase (GPx) and catalase (CAT) [9,41,52] that can protect against Cd-induced oxidative stress. Thereafter, the CC_50_ of both cells was selected for testing the anti-Cd toxicity property of CDE. 

### 3.5. Anti-Cd Toxicity of CDE in Cell Lines

Effects of the CDE treatment against Cd toxicity are shown in Figure 3. The results indicate that CDE protected and aided recovery of the cells in pre- and post-treatments (Figure 3B,C) but not in the combined-treatment (Figure 3A). Generally, Cd-induced toxicity uses several apoptotic pathways, such as disruption of the cellular antioxidant enzymes, inhibition of cell adhesion, induction of ROS and inhibition of DNA repair [5,6,7]. In fact, Cd is claimed to be a critical cause of imbalance between pro-oxidant and antioxidant homeostasis via oxidative stress, by catalyzing the formation of ROS, increasing lipid peroxidation, reducing glutathione, and reducing protein-bound sulfhydryl groups [43,61]. 

Prior results [15,46] confirm that CDE has high contents of phenolics and flavonoids, and should increase antioxidant enzyme activities of CAT and GPx. It is pointed out that CDE can protect and/or recover injured cells from toxic exposure to Cd. [62] reported that antioxidant enzymes act as ROS inhibitors. An increase in both antioxidants including antioxidant enzymes and exogenous antioxidant via polyphenols in the CDE improved the anti-Cd property. This result is in agreement with prior reports that aqueous CDE of the dried leaves significantly prevented kidney damage induced by Cd exposure in Wistar rats [63] and decreased the abnormal appearance and behavior of rat when administrated prior to Cd exposure [64]. However, the CDE seemed to not help cell survival in combined-treatment. This may due to the Cd-flavonoids interaction, led to be easy form to penetrate into cell membrane and affect to unbalance on cell because when the membrane leaks it has lost its barrier functionality. Generally, flavonoids can form complexes with metal cations via chelation [65] and the complexation of flavonoids with metal cations can considerably change their lipophilicity and interaction with the lipid bilayer, making it easier to penetrate into the hydrophobic sites of membranes [66]. Therefore, the flavonoid-Cd complex would get through into the cell and may be lysed by some enzyme in the cell, leading to free Cd and inducing cell death with acute toxicity. [67] reported that some lysosomal enzymes such as β-glucuronidase found in liver and kidney cells can play a crucial role in the cellular metabolism of flavonoids by releasing active flavonoids in the cell. Therefore, cell deaths were up in the combined-treatment compared with the negative control, due to a combined cytotoxicity effect. 

### 3.6. Anti-Cd Toxicity Property of Chelating Agent in the Test Cells

In this study, CaNa_2_EDTA was used as a chelating agent and as a positive control (Figure 4). The results show that a chelating agent can help cell survival in combined-treatment conditions for both cell types, HEK293 and HepG2 (Figure 4A). However, the chelating agent seemed to help cell survival not by much in pre- and post-treatments (Figure 4B,C), when compared with combined-treatment. The cytotoxicity with CDE and CaNa_2_EDTA in combined-treatment indicate that the active compounds in CDE may not use the same pathway as CaNa_2_EDTA effect did. In fact, CaNa_2_EDTA is a derivative of ethylenediamine tetraacetic acid (EDTA) and the main function of it is binding Cd ions to form complex ring-like structures or chelates, and removal of Cd toxic metal from the desired site in the body [68]. This chelating property can decrease Cd toxins and help cell survival, particularly in combined-treatment when compared with pre- and post-treatments, which were treated with CaNa_2_EDTA before and after Cd agent for 24 h respectively. Actually, CaNa_2_EDTA is the most commonly used chelating agent for heavy metal exposures, especially for lead and cadmium [68,69]. However, as is well-known CaNa_2_EDTA does not have a specific activity for chelating only Cd, but also for other essential minerals such as Zn, Cu, Fe, Co and Mn [70] involved in various enzyme functions as cofactors. Moreover, risks associated with CaNa_2_EDTA therapy have been reported, including bone marrow depression, prolonged bleeding time, convulsions, respiratory arrest, and renal failures [71]. Furthermore, some studies have reported that EDTA has cytotoxic effects and inflammation responses in animal cells [72,73]. In this present work, the cell viability of HEK293 was smaller than that of HepG2. It is well-known that HEK293 kidney cells have homeostasis mechanisms related to Ca^2+^ content [60]. The excess Ca^2+^ from CaNa_2_EDTA dissociation may lead to Ca^2+^ toxicity, reducing the viability of HEK293 cell line. This is similar to the finding in [43] suggesting that the Ca salt of EDTA provided major toxic side effects in the renal system, causing necrosis of tubular cells. 

The results indicate that using CaNa_2_EDTA in a combined-treatment was the most effective therapy. Therefore, the best concentration of CaNa_2_EDTA for each cell type in combined-treatment (100 µg/mL) was selected for the next experiments, studying combined therapy with the CDE against Cd toxicity. 

### 3.7. Anti-Cd Toxicity Property of CDE and Chelating Agent on the Test Cells

To study the interaction between the CDE and chelating agent, the experiment included 2 groups; pre- and post-treatments. Cd agent mixed with CaNa_2_EDTA was used to treat the cell lines before, and after treatment was with the CDE. The most effective concentration of CaNa_2_EDTA for each cell type in the previous part was 100 μg/mL for HEK293 and HepG2, and this was used as positive control. The results showed that cell viability significantly increased when the concentration of CDE increased, compared to negative control (Cd at CC_50_) and to positive control (Cd at CC_50_ mixed with CaNa_2_EDTA at 100 µg/mL). This indicates that the bioactive compounds in CDE can help and support CaNa_2_EDTA against Cd toxin via antioxidant properties of phenolic and flavonoid compounds. Moreover, the results show that cell viability of HepG2 was higher when CDE was used at 500–1000 μg/mL in the pre-treatment (Figure 5A) and at 1000 μg/mL in the post-treatment (Figure 5B) compared with the normal control, respectively. This indicates that the CDE may help cell proliferation or strength by some nutrition or bioactive compounds, such as glucoside and phenolic compounds, which were claimed to be supplementary compound supporting cell proliferation and protecting against cell death via antioxidant and anti-inflammatory effects [18,63]. Moreover, there were abnormal cell shapes, loss of adhesion, and cell merging when the cells were treated with Cd (Figure 6), and CDE was better at preventing or recovering from these abnormal phenomena than the negative and positive controls. This indicates that aqueous CDE of *T. laurifolia* leaves can be used as an herbal tea that can protect against Cd-induces oxidative stress. However, further study is necessary to clarify the *T. laurifolia* extract effects in an animal model, before clinical tests in human subjects.

## 4. Conclusions

Dried leaves of *T. laurifolia* contained protein, fat, ash, fiber, carbohydrate and some minerals; particularly K, P, and Mg. The present findings demonstrated that the aqueous extract of *T. larifolia* leaves significantly increased antioxidant enzyme activities in HEK293 kidney and HepG2 liver cells and significantly decreased MDA levels, in vitro. The extract could protect cells against Cd-induced toxicity for both cell types tested. The CaNa_2_EDTA agent can help cell survival better in a combined treatment than alone. Moreover, anti-Cd toxicity was higher when CDE and CaNa_2_EDTA were used together. Therefore, *in vivo* testing would be warranted before clinical tests in human subjects.

## Figures and Tables

**Figure 1 antioxidants-09-00047-f001:**
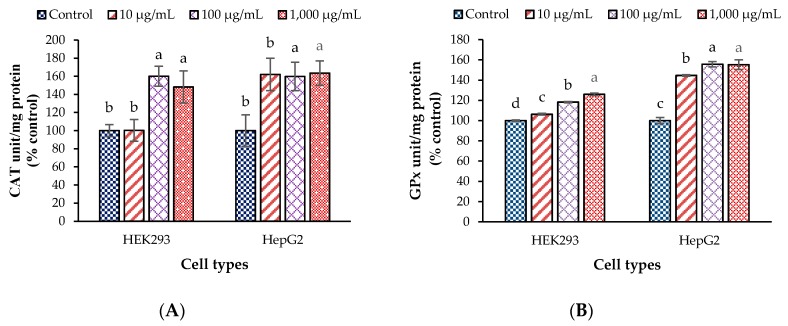
Antioxidant enzyme activities and malondialdehyde values of endogenous cellular extract (ECE) treated with *T. lauriforlia* extract in HEK293 and HepG2 cells; (**A**) catalase; (**B**) glutathione peroxidase and (**C**) malondialdehyde; ^a–d^ means with different letters are significantly different within a cell type (*p* < 0.05); Values are presented as mean ± standard deviation.

**Figure 2 antioxidants-09-00047-f002:**
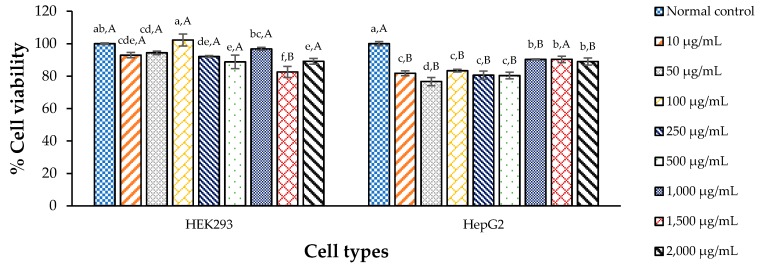
Of the CDE of *T. laurifolia* leaves on cell viability of HEK293 and HepG2 cells; ^a–f^ means with different letters are significantly different within a cell type (*p* < 0.05); ^A,B^ means within an extract concentration with different letters are significantly different (*p* < 0.05); Values are mean ± standard deviation.

**Figure 3 antioxidants-09-00047-f003:**
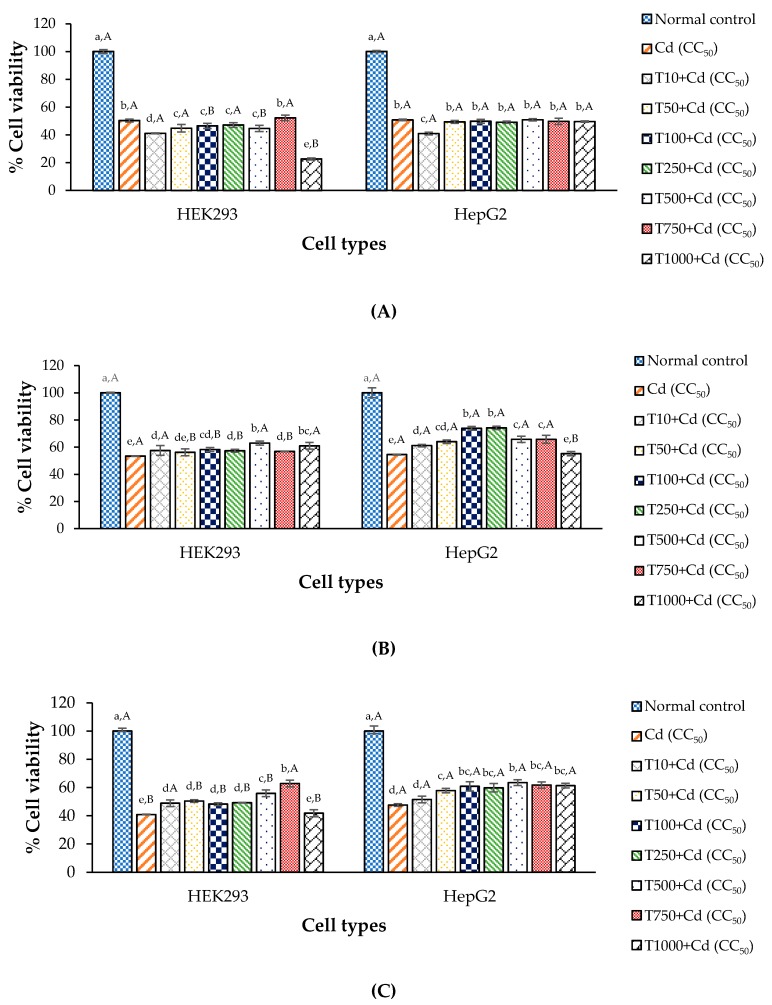
Anti-Cd toxicity of the CDE of *T. laurifolia* leaves in HEK293 and HepG2 cells determined by MTT assay; (**A**) combined-treatment; (**B**) pre-treatment and (**C**) post-treatment with CDE; ^a–e^ means with different letters are significantly different within a cell type (*p* < 0.05); ^A,B^ means within an extract concentration with different letters are significantly different (*p* < 0.05). T means the CDE of *T. laurifolia* leaves (μg/mL); Values are mean ± standard deviation.

**Figure 4 antioxidants-09-00047-f004:**
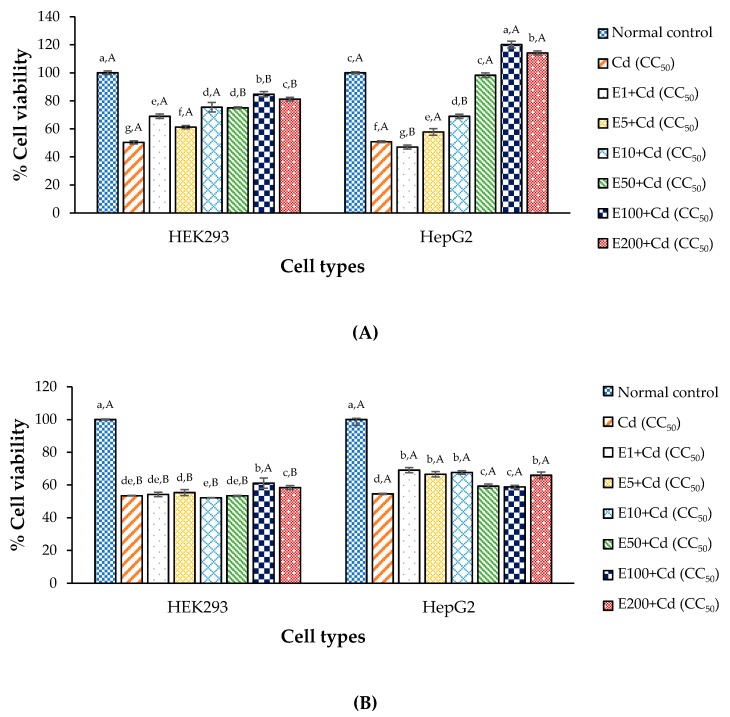
Anti-Cd toxicity of CaNa_2_EDTA in HEK293 and HepG2 cells determined by MTT assay; (**A**) combined-treatments; (**B**) pre-treatment and (**C**) post-treatment with CDE; ^a–g^ means with different letters are significantly different within a cell type (*p* < 0.05); ^A,B^ means within an extract concentration with different letters are significantly different (*p* < 0.05); E means CaNa_2_EDTA solution (μg/mL); Values are mean ± standard deviation.

**Figure 5 antioxidants-09-00047-f005:**
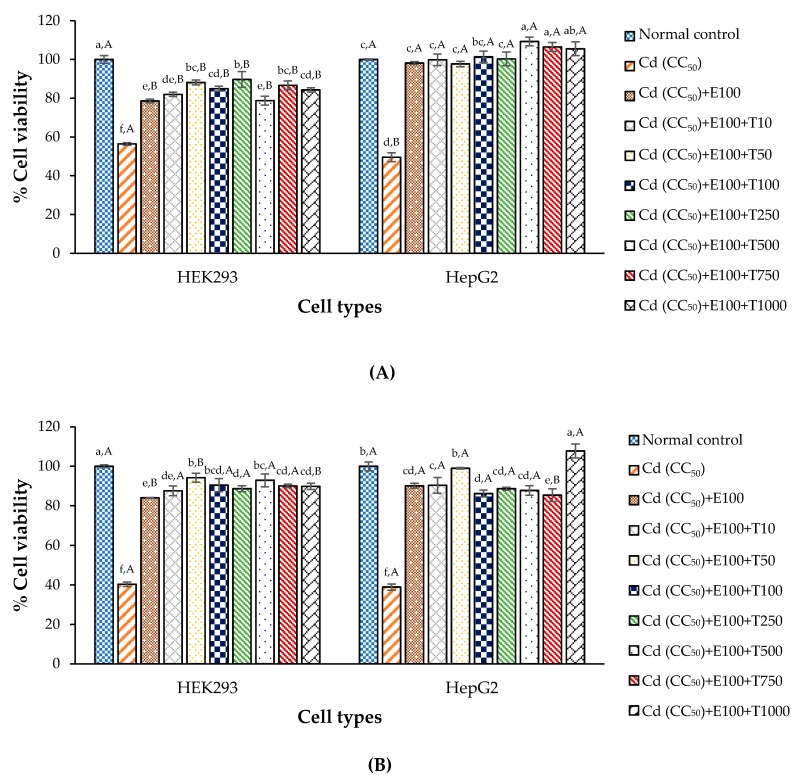
Anti-Cd toxicity of the CDE of *T. laurifolia* leaves coupled with CaNa_2_EDTA based on cell viability in HEK293 and HepG2 cells determined by MTT assay; (**A**) pre-treatment and (**B**) post-treatment; ^a–f^ means within a figure with different letters are significantly different (*p* < 0.05); ^A,B^ means within an extract concentration with different letters are significantly different (*p* < 0.05); T means the CDE of *T. laurifolia* leaves (μg/mL) and E means CaNa_2_EDTA at 100 μg/mL; Values are mean ± standard deviation.

**Figure 6 antioxidants-09-00047-f006:**
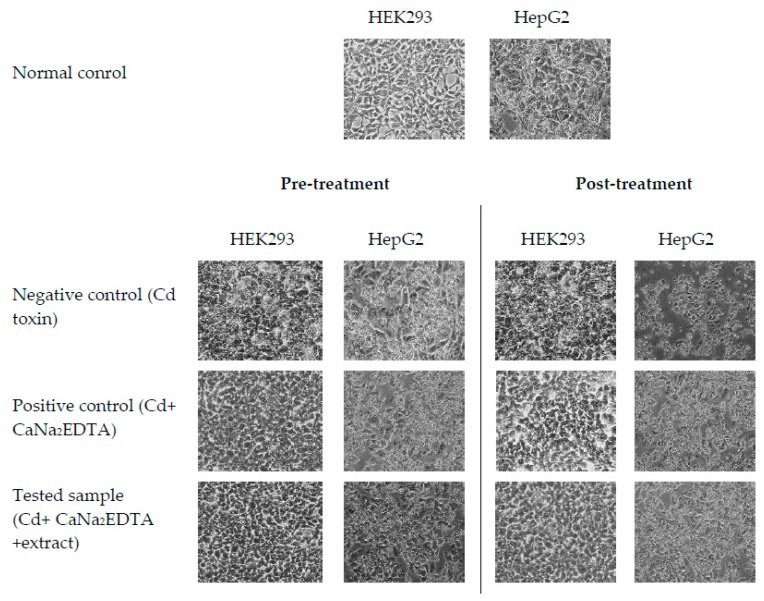
Morphology of HEK293 and HepG2 cells treated with Cd-induced toxicity and crude extract coupled with CaNa_2_EDTA in pre- and post-treatments.

**Table 1 antioxidants-09-00047-t001:** Proximate composition and mineral contents of *T. laurifolia* dried leaves.

Component	Dried-Leaf *T. laurifolia*
*Proximate compositions (% dry weight basis)*	
Crude protein	13.98 ± 1.90
Crude fat	1.83 ± 0.12
Ash	19.93 ± 1.90
Crude fiber	11.16 ± 0.44
Carbohydrate	53.10 ± 2.23
*Mineral contents (mg/kg dry leaf)*	
Potassium (K)	20,600.43
Phosphorus (P)	2500.75
Magnesium (Mg)	4548.30
Iron (Fe)	84.14
Zinc (Zn)	35.51
Copper (Cu)	16.99
Selenium (Se)	0.73

Values are presented as mean ± standard deviation.

**Table 2 antioxidants-09-00047-t002:** Cytotoxicity concentration at 50% (CC_50_) of CdCl_2_ from cells viability determined by MTT assay.

Cell Types	CC_50_ of Cd (μmol/L)
HEK293	64.09
HepG2	75.37

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
