# Peer review of "Thunbergia laurifolia Leaf Extract Increased Levels of Antioxidant Enzymes and Protected Human Cell-Lines In Vitro Against Cadmium"

_antioxidants, 2020, doi:10.3390/antiox9010047_

Round 1

Reviewer 1 Report

The present paper is interesting and well written but does not report particularly innovative results.

The authors determined the activity of CAT and GPx, both enzymes acting on hydrogen peroxide. Would it not be appropriate to also determine the activity of SOD, which dismutases the superoxide anion and produces hydrogen peroxide? We would have a more complete picture.

It would be necessary to explain why the Cd induces the formation of ROS, although it is not an active redox metal such as Cu and Fe.

In the results and discussion write the name of the species in italics.

Reviewer 2 Report

The manuscript presents an interesting work about the protective effect of an extract against the damage induced by cadmium. In my opinion, the work is well structured and procedures are clearly explained. Moreover, this is a well conducted research and I recommend its publication after minor revision. There are just several details that need to be checked before.

Page 1, lines 17-20: rewrite this sentence, it’s too long.

Page 1, line 37: I would recommend to check this affirmation. As far as I know there is a big difference in the toxicity found in liver and kidney in exposure to Cd. In liver, the presence of high amounts of metalotioneins help to avoid the toxicity of Cd in this tissue. Whereas kidney is a well-known target organ for Cd-toxicity.

For further studies in this field, I would recommend to measure ROS content and GSH content, they are very good biomarkers together with MDA. The changes in the antioxidant enzymes are sometimes difficut to explain without these complementary biomarkers.

Please check the abbreviations, for example in page 3, line 115 (DTNB), line 125 (TCA), line 126 (TBA)... I’m not sure they have been mentioned before.

Page 4, lines 139 and 155: I would suggest erasing these schemes, since they are not necessary to understand the procedure that has been previously explained.

In M&M section the procedure for the morphology study is needed.

Page 7, line 265: Please change the sentence to: “This is in agreement with Abdallah et al [53]…”

For further studies I would recommend using lower concentrations in order to detect sublethal effects such as oxidative stress. Maybe CC10, CC25… could be a better option than CC50.
